# Systematic Characterization of GATA Transcription Factors in *Liriodendron chinense* and Functional Validation in Abiotic Stresses

**DOI:** 10.3390/plants12122349

**Published:** 2023-06-16

**Authors:** Delight Hwarari, Yasmina Radani, Yuanlin Guan, Jinhui Chen, Yang Liming

**Affiliations:** State Key Laboratory of Tree Genetics and Breeding, Co-Innovation Center for Sustainable Forestry in Southern China, Nanjing Forestry University, Nanjing 210037, China

**Keywords:** *Liriodendron chinense*, GATA transcription factors, expression pattern, abiotic stress

## Abstract

The *Liriodendron chinense* in the *Magnoliaceae* family is an endangered tree species useful for its socio-economic and ecological benefits. Abiotic stresses (cold, heat, and drought stress), among other factors, affect its growth, development, and distribution. However, GATA transcription factors (TFs) respond to various abiotic stresses and play a significant role in plant acclimatization to abiotic stresses. To determine the function of GATA TFs in *L. chinense*, we investigated the GATA genes in the genome of *L. chinense*. In this study, a total of 18 GATA genes were identified, which were randomly distributed on 12 of the total 17 chromosomes. These GATA genes clustered together in four separate groups based on their phylogenetic relationships, gene structures, and domain conservation arrangements. Detailed interspecies phylogenetic analyses of the GATA gene family demonstrated a conservation of the GATAs and a probable diversification that prompted gene diversification in plant species. In addition, the LcGATA gene family was shown to be evolutionarily closer to that of *O. sativa*, giving an insight into the possible LcGATA gene functions. Investigations of LcGATA gene duplication showed four gene duplicate pairs by the segmental duplication event, and these genes were a result of strong purified selection. Analysis of the *cis*-regulatory elements demonstrated a significant representation of the abiotic stress elements in the promoter regions of the LcGATA genes. Additional gene expressions through transcriptome and qPCR analyses revealed a significant upregulation of LcGATA17, and LcGATA18 in various stresses, including heat, cold, and drought stress in all time points analyzed. We concluded that the LcGATA genes play a pivotal role in regulating abiotic stress in *L. chinense*. In summary, our results provide new insights into understanding of the LcGATA gene family and their regulatory functions during abiotic stresses.

## 1. Introduction

The *Liriodendron* genus encompasses two prominent species (spp.), *Liriodendron chinense* (Lc) and *Liriodendron tulipifera* [1]. Both are distinguished from annual herbaceous model plants, such as *Arabidopsis thaliana* and *Oryza sativa*, on account of their woody secondary growth and perennial habit [1,2]. The characteristics of the *Liriodendron* sp. define its commercial wood value, landscaping value, and timber supply [2]. Nonetheless, many factors affect the growth and development of *Liriodendron chinense,* including nutrient utilization, photosynthetic efficiency, hormonal regulation, and biotic and abiotic stresses [3].

Accumulating research has shown the involvement of GATA transcription factors (TFs) in regulating diverse growth processes and environmental responses in human and plant cells [4,5]. Members of the GATA gene family are evolutionarily conserved transcription factors (TFs) that exist in a wide range of organisms, ranging from cellular slime mold to vertebrates, plants, fungi, nematodes, insects, and echinoderms [6,7]. The GATA transcription factors comprise a protein family containing one or two highly conserved zinc finger DNA-binding domains, constituted by a type IV zinc finger in the form CX_2_CX_17–20_CX_2_C [7]. GATA proteins in animals are distinguished by a 17–18 residue in the binding loop, characterized by two CX_2_CX_17_CX_2_C zinc domain factors, with only the C-terminal involved in DNA binding [6]. Fungal GATA factors contain a single zinc finger domain and are categorized into two groups: the 17-residue loop (CX2CX17CX2C, also known as zinc finger type IVa) and the 18-residue loop (CX_2_CX_18_CX_2_C, also known as zinc finger type IVb) [8]. Previous studies have shown that most plants carry a single CX_2_CX_18_CX_2_C domain, with a few other plants containing one or two CX_2_CX_20_CX_2_C zinc finger domains [9,10]. In addition, the side chains of the zinc-finger GATAs in *A. nidulans* make hydrophobic contacts in the major groove of the DNA, while the carboxy-terminal tail contacts the phosphate backbone in the minor groove in the cGATA1 structure [9].

Previous studies have also evidenced several roles of the GATA transcription factors through binding to the consensus DNA sequence (A/T)GATA(A/G) during gene transcription regulation. Fungal GATAs, a combination of both the animal and plant zinc-finger domains, regarding amino acid residues present, act as inhibitors or regulators of transcriptional activities to downstream genes in metabolic processes, cell differentiation and plant development, nitrogen metabolism [11], circadian regulation [12], siderophore production [13], and mating-type switching [14]. In plants, a poplar gene, PdGATA19/PdGNC, is involved in photosynthetic electron transfer and carbon assimilation in the leaf, cell division, and utilization of carbohydrates in the stem through regulating hormonal responses [4]. Earlier studies in *A. thaliana* have shown the At5g56860 GNC to regulate carbon and nitrogen metabolism [10]. AtGATA2 has been identified as a critical transcriptional activator that mediates crosstalk between BR- and light-signaling pathways [15]. The LLM-domain and B-GATA transcription factors have been demonstrated to control greening in Arabidopsis [16].

Relating to this research, OsGATA16 in rice was shown to confer cold tolerance by repressing OsWRKY45-1 at the seedling stage [17]. OsGATA23a was shown as a multi-stress-responsive TF, showing elevated expression levels in response to salinity and drought [18]. Additional genetic, molecular, and physiological analyses of OsGATA8 have demonstrated its stress regulation roles in abiotic and biotic stress that result in yield increase achieved through regulating the expression of critical genes involved in stress tolerance, scavenging of ROS, and chlorophyll biosynthesis [19]. Additional analysis evidenced that OsGATA8 regulates salinity stress and drought by regulating the expression of critical genes involved in stress tolerance and scavenging of ROS [19]. Recently, Wang et al. [20] have shown that the overexpression of SlGATA17 in tomato enhances salinity tolerance through interacting with the SlHY5, coupled with the expression of other stress-related genes (SlDREB2A, SlAREB1, SlRD29A, and SlP5CS1), indicating that the GATA genes interact with other genes to significantly execute their biological functions. In other studies, two anthocyanins and GATA TFs (CGA1 and GNC) were shown to respond to different abiotic stresses triggered by ROS accumulations, thereby regulating the cold, heat, and drought stresses through interacting with other various downstream genes [21]. Again in Arabidopsis, GATA TFs CGA1 and GNC were shown to act as negative regulators of cold stress [22]. Nonetheless, further investigations showed that MADS-box TF SOC1 represses CGA1 and GNC to increase cold stress resistance [22]. BdGATA13 from *Brachypodium distachyon* was overexpressed in transgenic Arabidopsis, and the results showed transgenic wild plants that exhibited enhanced drought stress tolerance [23].

Several other functional GATA genes have been revealed, proving their importance in plant abiotic stress resistance and development; the GATA gene family has not yet been characterized in *L. chinense*. Cao et al. [1] reported on the reduced distribution of *L. chinense* in different time scenarios based on climate change and further predicted its extinction due to harsh weather conditions [24], suggesting the urgent need to increase the survivability of *L. chinense* in both the present and future time zones by enhancing its resistance to abiotic stresses. To explore the GATA gene family members in *L. chinense,* their composition, and possible biological processes, we analyzed the whole genome data of *L. chinense*. We systematically investigated the GATA gene family with regard to phylogeny, gene function differentiation, expression pattern, and stress responses. Our results showed a total of 18 LcGATA genes that were responsive to cold, heat, and drought stress. Generally, all the LcGATA genes showed differential expression patterns; notably, LcGATA18 exhibited high expression levels in all stresses analyzed in both the transcriptome and qPCR expression analyses, suggesting a pivotal role of LcGATAs in stress enhancement. Although this research is not exhaustive, it provides the basis for future gene functional research and genetic improvements in *L. chinense*.

## 2. Materials and Methods

### 2.1. Identification and Physicochemical Properties Analysis of GATA Family Members in Liriodendron chinense

The genomic and protein sequences of *Liriodendron chinense* were collected from the hardwood genomics database (now, TreeGenes database: https://treegenesdb.org/, accessed on 12 June 2022), following previous publications [25,26]. GATA protein sequences of *A. thaliana* were obtained from the TAIR database (https://www.arabidopsis.org/, accessed on 13 June 2022) [27]; protein sequences of *O. sativa*, *P. trichocarpa*, *M. polymorpha*, and *A. trichopoda* were retrieved from the online plant transcription factor database (http://planttfdb.cbi.pku.edu.cn, accessed on 13 June 2022) [28]. The GATA protein sequences of *A. thaliana* and *O. sativa* were used as query sequences against the *L. chinense* database using the local BLASTP program. The candidate proteins carrying the GATA domain were further screened. To authenticate the GATA protein candidates, the hidden Markov model (HMM) files of the GATA domain (PF00320) were obtained from the Pfam website (https://pfam.xfam.org/, accessed on 13 June 2022). An HMM search was performed on the *L. chinense* genome files, and GATA candidate genes were screened with E < 10^−5^ as the threshold value. The NCBI Batch-CDD search (https://www.ncbi.nlm.nih.gov/Structure/bwrpsb/bwrpsb.cgi, accessed on 13 June 2022) was used to confirm the identified candidates. The Molecular weight (Mw) prediction of GATA protein in *L. chinense* was computed by Compute-pI/Mw, and the Isoelectric Point Prediction (pI) was computed by the Expassy tool (http://web.expasy.org/Compute-pI/, accessed on 13 June 2022).

### 2.2. Phylogenetic Tree, Conservation Domain, Cis-Element, and Protein-to-Protein Analyses

Multiple sequence alignments were performed on the full-length amino acid sequences of GATA proteins in *L. chinense*, *A. thaliana*, *O sativa*, *P. trichocarpa*, and *M. polymorpha* with the MUSCLE program using default parameters as implemented in MEGA X [29]. Subsequently, MEGA X software (https://www.megasoftware.net/, accessed on 23 February 2023) was used to construct a phylogenetic tree based on the alignments using the neighbor-joining tree (NJT) method. All identified LcGATA proteins were predicted for subcellular localization using the DeepLoc-2.0 tool (https://services.healthtech.dtu.dk/services/DeepLoc-2.0/, accessed on 27 April 2023) [30]. The conserved domain (CDD), motif number and arrangements, and *cis*-elements were analyzed using previously described methods [31,32]. To gain knowledge of the protein interactions, we constructed a protein-to-protein network using the String Database (http://string-db.org, accessed on 14 June 2022). Full-length protein sequences of the LcGATA protein family were searched against the STRING database [33], and *Arabidopsis thaliana* was used as a reference plant species. Obtained tsv. format results were then transferred to Cytoscape software (https://cytoscape.org/, accessed on 23 February 2023) for protein network construction [34].

### 2.3. Plant Material Treatment and Transcriptome Data

The *L. chinense* somatic embryo-fetal regenerated seedlings from a single batch of constant growth were separated into three different incubators with different treatments: 4 °C for the cold stress treatment, 35–40 °C for the heat treatment, and 40% polyethylene glycol/PEG6000 for the drought stress. Mature leaves were collected at 0 h (control), 6 h, 1 d, and 3 d post-treatment and then stored at −80 °C for further use [26]. Three replicates were set for each independent experiment.

#### 2.3.1. RNA Quantification and Qualification

In addition, transcriptome sequencing was performed on the above sample. Total RNA was extracted from the leaves using the RNeasy Plant Mini Kit (Qiagen, Hilden, Germany) according to the manufacturer’s instruction. RNA degradation and contamination were monitored on 1% agarose gels. RNA purity was checked using the Nanophotometer^®^ spectrophotometer (IMPLEN, CA, USA). RNA integrity was assessed using the RNA Nano 6000 Assay Kit of the Bioanalyzer 2100 system (Agilent Technologies, Santa Clara, CA, USA). The RNA integrity was assessed using the RNA Nano 6000 Assay Kit (Agilent Technologies, Santa Clara, CA, USA). A total amount of 1 μg RNA per sample was used as input material for additional research [31].

#### 2.3.2. Library Preparation for Transcriptome Sequencing

Sequencing libraries were generated using NEBNext^®^ UltraTM RNA Library Prep Kit for Illumina^®^ (NEB Labs Inc., IPSWICH, Essex, MA, USA) following the manufacturer’s recommendations, and index codes were added to attribute sequences for each sample. Briefly, mRNA was purified from total RNA using poly-T oligo-attached magnetic beads. Fragmentation was carried out using divalent cations under elevated temperature in NEBNext First Strand Synthesis Reaction Buffer (5X). First-strand cDNA was synthesized using a random hexamer primer and M-MuLV Reverse Transcriptase (RNase H-). Second-strand cDNA synthesis was subsequently performed using DNA Polymerase I and RNase H [26].

The remaining overhangs were converted into blunt ends via exonuclease/polymerase activities. After the adenylation of 3′ ends of DNA fragments, a NEBNext Adaptor with a hairpin loop structure was ligated to prepare for hybridization. To select cDNA fragments of preferentially 250~300 bp in length, the library fragments were purified with the AMPure XP system (Beckman Coulter, Indianapolis, IN, USA). Then, 3 μL of USER Enzyme (NEB Labs Inc., IPSWICH, Essex, MA, USA) was used with size-selected, adaptor-ligated cDNA at 37 °C for 15 min, followed by 5 min at 95 °C before PCR. Then, PCR was performed with Phusion High-Fidelity DNA polymerase, Universal PCR primers, and Index (X) Primer. Finally, PCR products were purified (AMPure XP system; Beckman Coulter, Indianapolis, IN, USA) and library quality was assessed on the Agilent Bioanalyzer 2100 system, and transcriptome data were obtained.

The expression patterns of each LcGATA at each time interval were recorded and transformed to log base 2, and the expression level values were normalized to the maximum expression value. The expression pattern result was then displayed as a heatmap using the TBtools software (https://bio.tools/tbtools, accessed on 23 February 2023) [35].

### 2.4. qRT-PCR Expression Analysis

qRT-PCR analysis was used to authenticate the findings from transcriptome analysis, analyzing the expression patterns of LcGATA genes under three treatments of both the treated and control plants [36]. Samples from the above description were used. The KK-rapid plant total RNA extraction kit (Beijing Zoman Biotechnology Co., Ltd., Beijing, China) was used for total RNA extraction. The first-strand cDNA was synthesized with obtained RNA and an Evo M-MLV RT kit (GDNA. CLEAN for QPCRII AG 11,711 (Accurate Biotechnology Co., Ltd., Hunan, China). qPCR was performed using SYBR-green in the Roche LightCycler^®^480 real-time PCR system (Solna, Sweden). The relative expression levels of the genes were calculated with the ∆∆CT method. In addition, 18s rRNA was used as the internal reference [37]. All qRT-PCR primers were designed by Primer5.0 and are listed in Appendix A. Additionally, before qRT-PCR analysis, all the primers were verified for primer specificity using the PCR and viewed using gel agarose electrophoresis. The qRT-PCR result was displayed using GraphPad Prism and Microsoft Excel [38].

## 3. Results

### 3.1. Characterization of GATA Factors in L. chinense and Their Physiological Properties

BLASTP search was used to mine potential GATA TFs from the *L. chinense* (Lc) genome, and the identified LcGATA proteins were authenticated for the zinc finger domain (PF00320) using the HAMMER3.1. In total, 18 LcGATA protein sequences were yielded after removing redundant sequences and duplicates. Comparably, *L. chinense* had relatively fewer GATA factors than Arabidopsis and rice, with 30 and 29, respectively [9]. To further analyze the sequence features of the 18 LcGATA proteins, their amino acid sequences were aligned using Mega X (Figure 1 and Appendix A). The alignment result revealed that most LcGATA proteins contained only a single zinc finger domain, CX_2_CX_18_CX_2_, including LcGATA1, LcGATA2, LcGATA3, LcGATA6, LcGATA7, LcGATA9, LcGATA10, LcGATA11, LcGATA12, and LcGATA16; two proteins, LcGATA4 and LcGATA8, carried a partial zinc finger domain, suggesting that they had lost some amino acids. The remaining LcGATAs carried a single zinc finger domain, CX_2_CX_20_CX_2_ (Figure 1). In addition, the LcGATAs were grouped into Groups A to D according to their protein alignments and previous publications [15,18].

To determine the chromosomal location of the LcGATA genes, the TBtools software (https://bio.tools/tbtools, accessed on 23 February 2023) was used to map the identified LcGATAs on the *L. chinense* genome (Figure 2). A total of 18 LcGATAs were mapped onto 12 chromosomes (chrs); chr 1 contained 3 genes, accounting for 16.7% of the total LcGATAs. In total, 3 chromosomes, chr 3, 6, and 11, had 2 LcGATA genes each, and the rest had 1 LcGATA gene each. Based on their chromosomal locations, the identified LcGATAs were renamed LcGATA1-18. Their physiochemical properties were computed to better understand the LcGATAs (Table 1). The lengths of the *L. chinense* GATAs varied from 120 (LcGATA11) to 624 (LcGATA14) amino acids (aas.). The isoelectric point of the LcGATAs ranged from 4.63 (LcGATA8) to 10.05 (LcGATA16), indicating that the LcGATAs ranged from weakly acidic to weakly basic. The subcellular predictions confirmed that all the LcGATA proteins were localized in the nucleus except for LcGATA8, which was localized in the cytoplasm.

### 3.2. Phylogenetic Analysis of GATA Factors

The systematic classification of the transcription factor families based on molecular phylogeny facilitates the understanding of functional and genomic studies [40]. To better comprehend the evolutionary relationship of the LcGATA genes, an unrooted phylogenetic tree was generated using the neighbor-joining tree (NJT) method. A total of 134 proteins, including those from previously published GATA findings in model plants, were considered, that is 18 LcGATAs, 30 AtGATAs, 29 OsGATAs, 39 PtGATAs, 6 MapolyGATAs, and 19 VvGATAs (Figure 3). The protein sequences clustered randomly into four groups based on protein structures, domain arrangements, and probable functional similarities. The cluster groups were named A to D, following previous publications [41,42]. In-depth analysis showed huge divergences, although group D was marked with less divergence, having the least number of GATA proteins. The other groups showed a divergence from the same monophyly branch, and we hypothesized that groups A and B were the last to diverge based on their positions. We also observed minor solo subgroups within the major annotated groups, although these were not named (Figure 3). Conclusions were drawn that the GATA proteins divided into various groups and subgroups based on protein similarities and other factors. Interestingly, the *L. chinense* GATA protein family was fully represented in each group, also suggesting a wider functional range of the LcGATAs. In summary, these findings show that the GATA TFs exhibit diverse biological functions and are conserved in plants.

To gain insight into the evolutionary relationship of the plant species, we constructed an interspecies phylogenetic tree using orthologs determined by the Xshell ortho-finder software (https://xshell.en.softonic.com/, accessed on 23 February 2023) (Figure 4A) [43]. We observed that the plant species had a common ancestor and diverged through speciation. Specifically, *Marchantia polymorpha* diverged earlier than the angiosperms, suggesting that the wort has a highly conserved genome base. However, divergence was more significant in the angiosperm clade, forming different plant species through evolution. Thence, we concluded that expansion in the angiosperm clade may have resulted in gene function gain or loss during adaptation.

Specific group analysis showed that groups A, B, C, and D contained 55, 38, 26, and 15 GATA protein sequences, respectively (Figure 4B). Groups A (58) and D (15) carried the most and the least GATA protein sequences among the categorized groups. Plant order comparisons exhibited that the angiosperms had the most GATAs compared to other plant orders, which we related to whole-genome duplication events in angiosperms. For instance, *P. trichocarpa*, an angiosperm, and *M. polymorpha*, a liverwort, had 39 and 6 GATA proteins, respectively.

### 3.3. The Conserved Domain, Motif, and Gene Structure Analysis

Protein sequence motifs are protein signatures that better aid in predicting protein functions. To better understand the LcGATA protein motif arrangement, we used the online tool MEME. In this analysis, the LcGATA protein sequences clustered together into three clusters (Cluster A to C) based on the motif arrangement similarities (Figure 5A). A total of 15 motifs were identified and designated as motifs 1 to 15 (Figure 5B and Appendix A). Motif 1 was speculated to be the zinc finger domain since it was present in all the LcGATAs except LcGATA4. Nonetheless, the LcGATA4 carried the zinc finger superfamily domain. In-depth analysis showed that cluster A carried 2 to 7 motifs, including motifs 14, 3, 5, 1, and 13; carrying the most significant representation of LcGATA motifs identified. Cluster B had at least 4 motifs, and Cluster C carried at most 3 motifs. These results show that different LcGATA proteins exhibit different motif arrangements and possibly different functions.

The LcGATA protein conserved domain arrangements were established using the online NCBI-CDD. Likewise, the LcGATA proteins clustered according to similar CDD arrangements (Figure 5C). Notably, cluster A carried the most conserved domains, including the ZnF_GATA, CCT, and the abhydrolase superfamily domain. On the other hand, the rest of the LcGATA proteins carried a single domain, either a ZnF_GATA or the GATA domain. Interestingly, the LcGATA14 had the most significant number of conserved domains present, including a TCP domain, as evidenced previously [9], suggesting that the LcGATA TFs are also invested in other functions carried out by different TF families.

Gene structure analysis is crucial in understanding and predicting possible biological functions of genes [45]. Similarly, LcGATA genes clustered together in three groups based on their similarities in gene structure and arrangements (Figure 5D). Cluster A had two to three exons, and most of the genes had no introns present, except for LcGATA3, which had two introns flanking at both ends. Exceptionally, the LcGATA6 had both exons positioned at the C-terminal, and LcGATA16 had a single intron in the C-terminal. Cluster B had an exon range of six to seven, and LcGATA17 had the highest number of exons (seven). Additionally, cluster B had a consistent intron number of two, flanking the N- and C- terminals. Cluster C was characterized by at least two short exons and two introns flanking at both the N- and C- terminals. In summary, genes with the same structure and exon-intron distribution clustered, possibly suggesting similar biological functions of the clustered genes.

### 3.4. Collinearity, Selection Pressure, and Duplication Events

Gene expansion in *L. chinense* has been previously reported to be mainly by a single lineage-specific WGD event that occurred approximately 116 million years ago (m.y.a) [2]. Two evolutionary driving forces, tandem and segmental duplication, have been previously identified in various plant species [46]. Therefore, we performed synteny analysis to determine the expansion of the LcGATA genes in light of the tandem and segmental duplications (Figure 6). The results showed no tandem duplication and four pairs of segmentally duplicated genes (LcGATA7-LcGATA16, LcGATA11-LcGATA2, LcGATA18-LcGATA15, and LcGATA17-LcGATA14), suggesting that the segmental gene duplication event contributed to the LcGATA gene family generation and expansion.

Collinearity within interspecies may also reflect phylogeny. We used the TBtools software (https://bio.tools/tbtools, accessed on 23 February 2023) to generate and deduce the synteny analysis of the LcGATA gene family between three model plants, *A. thaliana*, *P. trichocarpa*, and *O. sativa.* The results reflected that *L. chinense* had more orthologous genes with *P. trichocarpa* than with *O. sativa* and *Arabidopsis thaliana* (Figure 7). In detail, 13, 10, and 6 orthologous gene pairs were detected between *P. trichocarpa*, *O. sativa*, and *A. thaliana* with *L. chinense*, respectively. Although *L. chinense* had more orthologous gene pairs with *P. trichocarpa,* it is evolutionarily closer to *O. sativa.* This finding suggests that collinear genes of *L. chinense* to other plant species may possess similar functions, thereby suggesting a possible way to find potential LcGATA gene functions.

The rate of nonsynonymous (Ka) to synonymous (Ks) substitution was used to evaluate the degree of the evolutionary change based on natural selection [47] in *L. chinense* (Table 2). In all the identified gene pairs, the calculated values of Ka were less than the Ks values, (Ka < Ks), and respective nonsynonymous to synonymous ratios (Ka/Ks) were below 1 (0.19–0.23), indicating a purifying or stabilizing selection of the LcGATA genes during the evolutionary process.

### 3.5. Protein-Protein Interaction and Secondary Structure of LcGATAs

Protein-to-protein interaction (PPI) analysis is crucial in elucidating protein function and the impact of protein absence or presence [48]. Therefore, we performed a PPI investigation to understand the LcGATA protein function (Figure 8A); using the STRING online database, LcGATA proteins were searched against their orthologous proteins in *A. thaliana*. The network showed a dense interconnection between the evolutionary GATA groups A to D. The results showed that more group C (colored in yellow) LcGATA proteins interacted with other GATA protein groups to perform their functional roles. Interestingly, LcGATA3 in group D (colored in purple) was positioned as a central hub to which several other proteins link, suggesting that LcGATA3 is involved in several functional roles. Additionally, LcGATA3 formed a direct linkage with LcGATA1.

Comprehension of the protein 3-dimensional (3D) structure is a huge hint in understanding protein function. We constructed the 3D protein structure of four proteins, LcGATA15, LcGATA16, LcGATA17, and LcGATA18, using the online Expassy SWISS-model (https://swissmodel.expasy.org/, accessed on 16 June 2022) (Figure 8B–E). The results showed 16 α-helices, 5 β-strands, and 12 coils in LcGATA15, with most of the β-strands exhibiting high thermal mobility as opposed to other regions. The LcGATA16 protein had four α-helices, three β-strands, and six coils; most of the coil strand residues had high thermal mobility, and most amino acids were buried residues. LcGATA18 had four α-helices, four β-strands, and eight coils; most of the coil strand residues had high thermal mobility, and the amino acids were lowly buried. These findings show that the investigated LcGATA proteins had different structures and chemical properties. Additionally, buried amino acid residues present in all the investigated proteins suggested hydrophobicity in those regions, evidencing that the LcGATAs interact with DNA for different functions.

### 3.6. Analysis of Cis-Acting Elements in Promoters of LcGATAs

To determine the mechanisms by which the GATA genes respond during transcriptional regulation, we analyzed the *cis*-acting elements in the promoter regions of the LcGATA genes using the online tool PlantCare [49] (Figure 9). Generally, the *cis*-acting elements were abundantly distributed in the promoter regions of LcGATAs, implying that the LcGATAs are involved in a wide range of functional processes. The predictions showed 482 *cis*-acting-type elements and other multiple elements involved in various functions. The observed *cis*-acting elements were classified into three groups according to their similarities: growth and development constituted 30.5% (147), abiotic and biotic responses constituted 42.3% (206), and phytohormone responses constituted 26.5% (129) (Figure 9A). An in-depth analysis showed that the six elements classified in the growth and development group were abundant in LcGATA16, LcGATA13, LcGATA8, LcGATA6, and so on (Figure 9A). The G-box and ARE elements were the most abundant in this group, totaling 59 and 45, respectively. Hence, it was insinuated that these genes might be involved in growth and development functions. In the phytohormone response, most *cis*-acting elements were present in the abscisic acid responsiveness (ABRE) elements, and LcGATA10, LcGATA11, and LcGATA9 had a huge representation of these elements in their promoter regions.

This research mainly focused on stress responses, so 10 elements were classified into the abiotic- and biotic-responsive groups (Figure 9B). Notably, STRE and MYB elements had the most enrichment of 54 and 73, respectively, among other *cis*-acting elements. While, the Drought-responsive 1 (DRE1) and MBS elements had the least enrichment, with only three elements present, detected in LcGATA10 and LcGATA13, and one element was detected in LcGATA13. In summary, we noted that all the LcGATA genes had more than one stress response element, implying that the LcGATAs are actively involved in stress responses. This result indicates that the GATA genes in *L. chinense* are engaged in a wide range of biological functions, including abiotic stress response.

### 3.7. Transcriptomic and qRT-PCR Gene Expression Analysis of LcGATAs under Abiotic Stresses

We performed a transcriptomic gene expression analysis to shed light on the potential biological functions of the LcGATA genes in three abiotic stresses (cold, heat, and drought stress) (Figure 10). Generally, the LcGATA genes were differentially expressed in different abiotic stresses. The individual LcGATA gene expressions clustered into three groups, according to their expression similarities. The first group was composed of six LcGATA genes (LcGATA 2/3/6/9/11/16), characterized by moderate to high expression patterns throughout the treatment period, suggesting that these genes actively respond to abiotic stress regulation. Nonetheless, LcGATA9 and LcGATA2 were fairly downregulated during cold stress at 1 d and 3 d, indicating that these two genes are repressed during cold stress at long time points. The second group consisted of five LcGATAs (LcGATA18/17/15/14/7) which similarly had moderate to high expression patterns from treatment onset to termination in all stresses investigated. Notably, LcGATA18 had the highest expression pattern compared to other genes. Furthermore, there was an insignificant downregulation expression pattern in LcGATA11 and LcGATA14 during the heat stress at 6 h, which was then upregulated at 1 d. The third group had six LcGATA genes (LcGATA1/5/8/10/12/13), which were lowly expressed and characterized by low upregulation and downregulation expression patterns. Specifically, LcGATA8 and LcGATA5 were fairly downregulated, except for LcGATA8 during the heat stress, which was later upregulated at 6 h and 1 d. This result shows that LcGATA8 and LcGATA5 genes are somewhat involved in abiotic stress regulation. In addition, LcGATA4 was highly downregulated throughout the treatment period in all stresses, suggesting that LcGATA4 is not involved in abiotic stress regulation.

To further authenticate our transcriptomic expression result, we selected nine LcGATA genes (LcGATA3/7/9/10/12/14/15/17/18) based on their response patterns in the transcriptome analysis for qPCR gene expression analysis (Figure 11). The result showed differential expression patterns that significantly differed for each time point and stress. Generally, the qPCR expression result was consistent with the findings of the transcriptomic gene expression.

Prompted by the results of gene expression analysis in different abiotic stresses, we compared the correlation of stress regulatory genes, the LcCBF transcriptome gene expression analysis shown previously [50], with the transcriptomic gene expression result of LcGATA genes (Appendix A). Generally, most of the LcGATA gene transcriptome expressions correlated with those of the LcCBF genes. Notably, LcGATA17 and LcGATA18 had a higher correlation factor with several LcCBFs, including LcCBF3-10, in both heat and cold stress. In summary, transcriptomic expression analysis of LcGATA correlated with that of LcCBF genes at specific times, supporting the fact that LcGATAs play a pivotal role during abiotic stresses.

## 4. Discussion

GATA transcription factors are evolutionarily conserved in animals, fungi, and plants; the total number of GATA factors in plants is increased compared to that in other organisms [51]. Due to their expanse in plants, they have been classified into several groups, which is essential for their functional diversity. To date, GATA transcription factors have been identified in various plants, such as *Phyllostachys edulis* [52], *Triticum aestivum* [53], *Gossypium hirsutum* [54], *Populus trichocarpa* [41], *Arabidopsis thaliana*, and *Oryza sativa* [9]. In addition, the functional roles of GATA factors have been elucidated in many plant species and are yet to be explored in the *L. chinense* plant species.

In this research, 18 LcGATA genes were identified in the genome of *L. chinense*, although this is fewer than the total numbers in Arabidopsis and rice (Figure 4B) [9]. This phenomenon can be related to the differences in gene duplication events, adaptation measures, and evolutionary events leading to varied gene family sizes [21,42,53]. The identified LcGATA genes were observed to carry a single zinc finger domain, CX_2_CX_18-20_CX_2_, which was also evident in the conserved domain and motif analyses (Figure 5). Reyes et al. [9] first demonstrated the zinc finger GATA domain as the DNA-binding region of plant GATA TFs, which associates with various factors that affect its functions [41,55]. In support of these findings, the physiochemical properties analysis showed multiple properties of the identified LcGATAs, implying an extended range of functionality. Other researchers have also demonstrated numerous functions of the GATA factor in improving postharvest sprouting in wheat through the TaCATA1-TaELF6-A1-TaABI5 module [56], drought resistance during long-term exposure of the HOVUSG2784400 [57], and so on. Investigation of the protein chemical structures and properties (Figure 8) also evidenced different forms and compositions. Notably, we noted that the investigated LcGATA proteins carried specific hydrophobic amino acid residues, alluding to the fact that GATA TF hydrophobic interactions determine their binding-site specificity [8].

Observed differences in the protein structure and arrangement prompted phylogenetic analysis. Based on protein arrangement similarities, the result confirmed the classification of LcGATAs into four groups (groups A–D) (Figure 3). However, previous research has shown the classification of GATA factors in seven groups (groups A–G) [8]. In support of our findings, some recent studies have also classified GATA TFs into four groups [5,17,23]. For instance, studies in poplar GATA factors have divided the GATA family into four groups and further hypothesized that this disparity in the total number of group classifications among species may be related to gene loss or gain before or during the species evolution of monocots and dicots [4]. In addition, phylogenetic analysis of GATA TFs showed that LcGATAs had full representations in each group, implying an extended range of LcGATA gene functionality. Some LcGATA genes even clustered in the same monophyletic group with *M. polymorpha* GATA proteins. Thus, the LcGATA can be arguably defined as conserved TFs, since the *Marchantia polymorpha* has become a model organism of choice that provides clues to mechanisms underlying eco–evo–devo biology in plants [58]. Similarly, LcGATA3 was in the same group as OsGATA16 in group B, suggesting that LcGATA3 may be involved in cold stress tolerance [17]; seven LcGATAs were classified in group A, also providing a probability that these genes may be involved in Floral organ boundaries and development, such as AtGATA3/4/2 [59]. Using phylogenetic analysis, we concluded that all GATA proteins evolved from a common ancestor that diversified into different plant species through speciation and specialized into diverse genomic functions [60]. Specifically, LcGATAs may be involved in various abiotic stress regulation roles.

Observed variations in GATA gene family size among plant species and total genome size have been attributed to genome or gene duplication events [60]. In this research, we investigated LcGATA gene duplications and collinearity (Figure 6). Our synteny analysis showed that the LcGATA gene factors had five genes paired together by segmental duplication, demonstrating that segmental duplication is the main factor contributing to the expansion of LcGATA genes. These findings are similar to research in *V. vinifera* [42] and *B. napus* [61]. Similarly, previous gene family studies in *L. chinense*, such as in TCP [26], SnRK [36], and bZIP [37] gene families, have all shown that segmental duplication might have been the driving force of gene family expansions in *L. chinense*. Therefore, we concluded that the segmental duplication event is also responsible for the expansion of the LcGATA gene family. Ka/Ks value computation showed Ka < Ks and Ka/Ks < 1, signifying a purifying selection. These findings suggest that the Ka genes (LcGATA10, LcGATA16, LcGATA18, LcGATA13, and LcGATA17) might be some of the genes behind how the *L. chinense* species adapts to external environmental stresses [31,62].

In this research, we also showed the involvement of LcGATA genes in abiotic stress regulation. Previously, GATA genes in wheat were demonstrated to respond to stress by linking the CBF genes, although the specific mechanism remains a mystery [56]. In this research, the *cis*-regulatory analysis result exhibited the presence of various *cis*-elements in growth and development, stress responses, and phytohormone-responsive factors, evidencing that LcGATAs might be involved in multiple biological functions. The stress-responsive elements were particularly of interest to this research; thus, we showed that more stress-responsive elements were present in LcGATA16, LcGATA17, and LcGATA18 among all the other LcGATA genes. Therefore, we concluded that these LcGATAs are actively involved in stress regulation. To support this finding, LcGATA3 clustered together with OsGATA16 in the phylogenetic analysis, suggesting a possibility that LcGATA3 might also regulate cold stress by repressing the WRKY genes in *L. chinense* [17]. Wu et al. [63] have also shown that the WRKY genes in *Liriodendron chinense* respond to cold stress; nevertheless, more research is still needed to show cold response mechanisms.

Abiotic stress responses of the LcGATA genes were investigated by transcriptomic expression analysis and qPCR gene expression validation. Outstandingly, LcGATA18 was highly upregulated, nine LcGATA genes (LcGATA2, LcGATA3, LcGATA6, LcGATA7, LcGATA9, LcGATA11, LcGATA14, LcGATA15, and LcGATA16) were moderately upregulated, and the remaining were significantly downregulated for all the three stresses, supporting the fact that LcGATA genes respond to abiotic stresses at varying extents and periods. In other studies, OsGATA23a has been shown as a multi-stress-responsive TF, showing elevated expression levels in response to drought and salinity [18]. Another study in *A. oryzae* has demonstrated that AoGATA genes are involved in drought and high-temperature regulation [64]. Therefore, these studies support the fact that LcGATA genes respond to abiotic stresses. Nonetheless, the mechanism is still unknown. CBF genes have been identified as cold-responsive genes [50]; we compared the transcriptomic correlation results of the LcCBF gene with those of the LcGATA genes. The results showed a strong correlation in most of the genes at the same time in similar stress treatments, especially in LcGATA17 and LcGATA18. Therefore, we postulated that LcGATA genes might regulate abiotic stresses, such as cold stress, by directly or indirectly linking with LcCBFs. Research has demonstrated that the LcGATA genes link with various TFs and other DNA modules to effect their full function [65,66,67]. In summary, this research evidenced that LcGATA genes are involved in abiotic stress regulation, especially LcGATA18. However, additional research is still required.

## 5. Conclusions

The characterization of GATA transcription factors in *Liriodendron chinense* enabled the elucidation of the functional conservation and divergence of LcGATAs. We identified 18 GATA genes in the genome of *L. chinense*. We analyzed their physiochemical properties, gene arrangements and structures, motifs and conserved domains, phylogeny and evolution, and gene expression patterns. We concluded that LcGATAs are evolutionary conserved and primitive genes that have expanded in the *L. chinense* genome through the segmental duplication event. The transcriptome and qRT-PCR gene expression analyses showed that LcGATA responds to cold, heat, and drought stresses. We concluded that LcGATAs possibly regulate stresses by linking with other transcription factors, such as CBFs and WRKYs. However, these results are inconclusive; they provide a basis for further functional research in *L. chinense* GATA factors. Future studies should focus on substantial expression analysis of the LcGATA genes and further bi-engineering their genetic makeup to enhance abiotic stress resistance and *L. chinense* survival and distribution.

## Figures and Tables

**Figure 1 plants-12-02349-f001:**
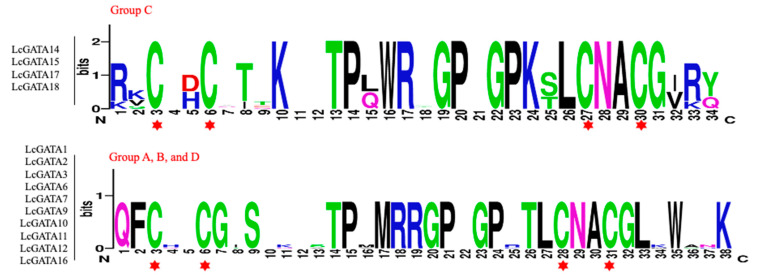
The zinc finger conserved domains in the *L. chinense* GATA protein sequences. The sequence logos were obtained using WebLogo online tool (http://weblogo.berkeley.edu/, accessed on 16 June 2022) [39] based on the protein alignments of the GATA domains. The overall height of each stack letter indicates the sequence conservation at that position (measured in bits and shown on the *y*-axis). Additionally, each stack letter reflects the relative frequency of the corresponding amino acid at that position, each amino acid position is denoted by numbers on the *x*-axis. The red star depicts 90% conserved loci within the whole family.

**Figure 2 plants-12-02349-f002:**
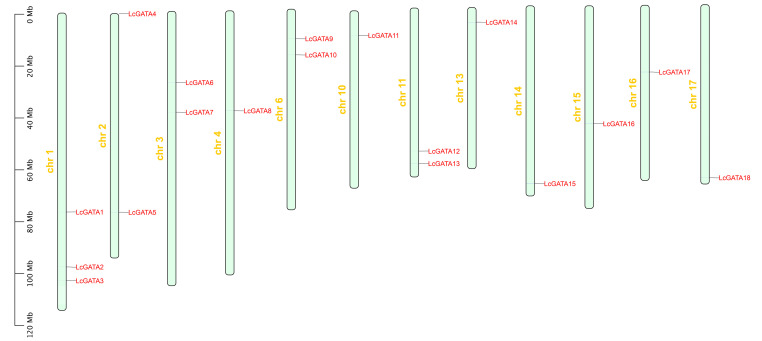
Characterization and location determination of *L. chinense* GATA genes, showing the location of 18 LcGATA genes on 12 *L. chinense* chromosomes (represented by light green blocks, and chromosome number is denoted in yellow). Each LcGATA gene is denoted in red font, the far-left scale shows the respective lengths of each chromosome.

**Figure 3 plants-12-02349-f003:**
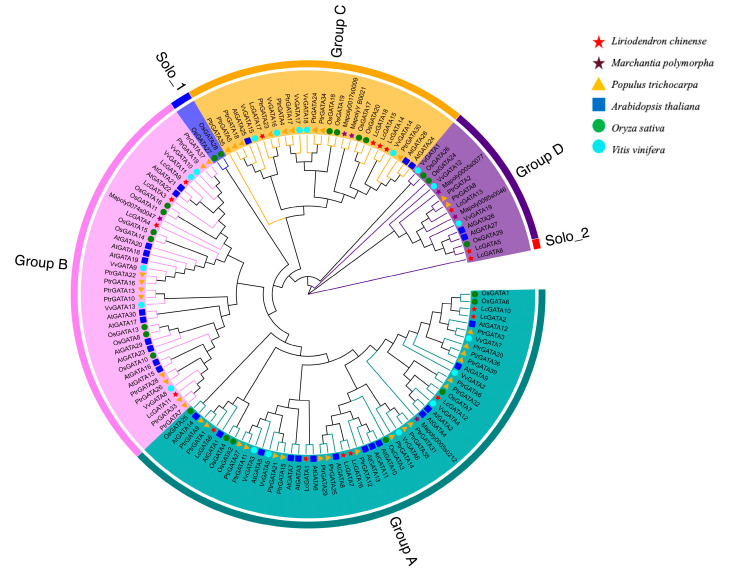
Phylogenetic analysis of the GATA transcription factors showing the evolution of 134 GATA protein sequences. The phylogenetic tree was generated using the neighbor-joining tree (NJT) method in Mega X, prior sequences were merged and aligned using MUSCLE in Mega X. Different color shapes represent different taxon names, as shown in the key (top right corner). Furthermore, GATA proteins were categorized into four groups (A to D) according to their clustering, shown with different color branches and boundaries.

**Figure 4 plants-12-02349-f004:**
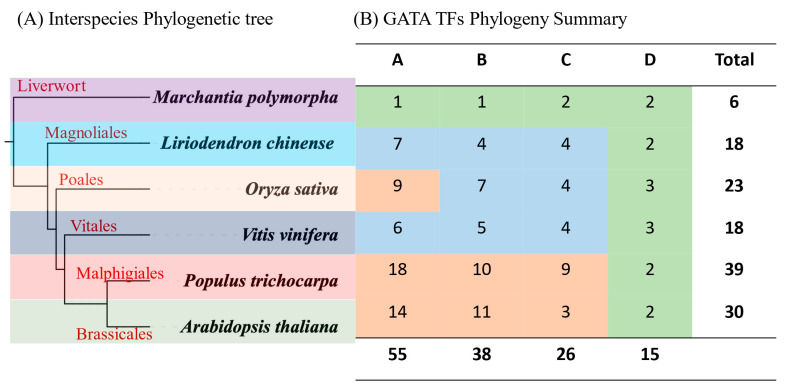
Plant order evolution. (**A**) The phylogenetic tree of different plants in different orders. The phylogenetic tree was constructed from selected plant orthologs generated by the Xshell software (https://xshell.en.softonic.com/, accessed on 23 February 2023) and the online tool iTOL [44]. Different color backgrounds show different plant orders. (**B**) A summary of the total number of GATA proteins present in each plant and group in Figure 3. The background heat map shows varying group sizes of LcGATA TFs in each plant analyzed. The green-to-red color change indicates a range from the lowest to the highest total number of GATA TFs.

**Figure 5 plants-12-02349-f005:**
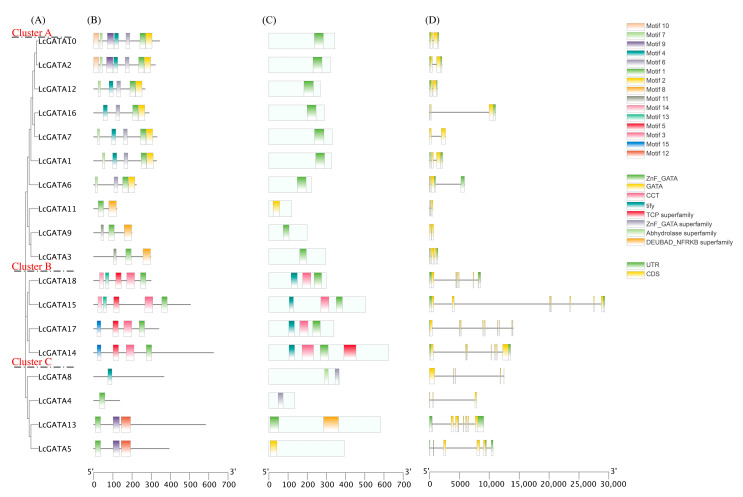
LcGATA protein motif structure and arrangement. (**A**) The phylogenetic tree of *Liriodendron chinense* GATA factors. LcGATA proteins were aligned, and the tree was constructed using the NJT method by MegaX at a bootstrap value of 100. Tree clusters were named A to C, shown on the far left. (**B**) The motif arrangement in different LcGATA proteins, analyzed using the MEME online tool. The motifs detected were numbered 1–15 (Appendix A); the key is outlined in the top right corner. (**C**) The conserved domain arrangements in different LcGATA proteins; different domains are shown in different colors, fully described in the key second bottom right corner. (**D**) The exon-intron number and arrangements of different LcGATA genes. The exons are depicted in yellow, while the introns are shown in green. The scale below indicates the approximated lengths of the protein structures.

**Figure 6 plants-12-02349-f006:**
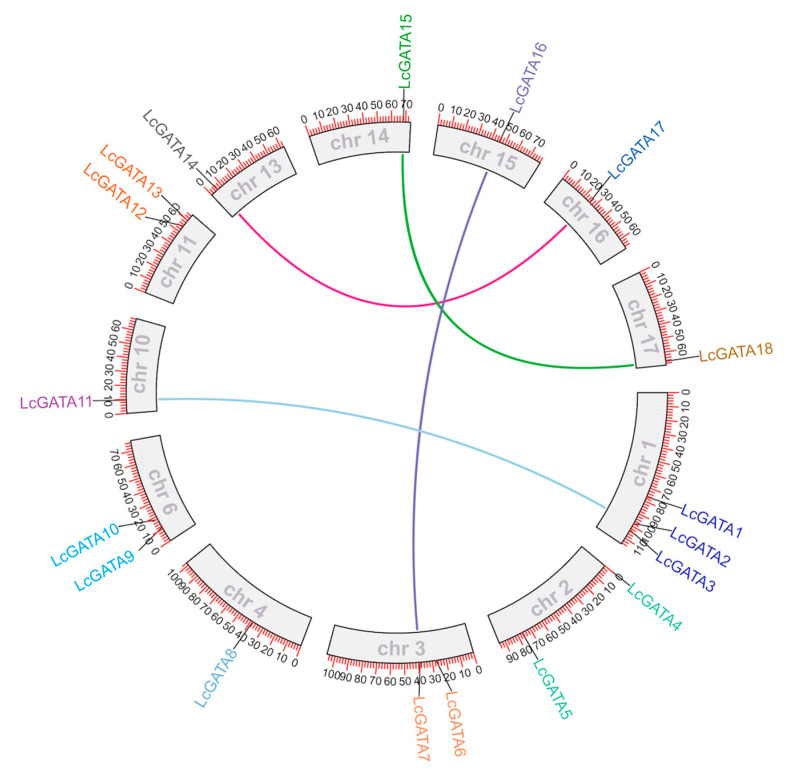
Gene collinearity analysis of the GATA genes in *L. chinense*. The approximate positions of LcGATAs are marked with a short red line on the circle outside the synteny blocks (presenting chromosomes), and their specific gene positions on each chromosome are denoted by numbers outside the chromosome blocks. Colored curves show the gene collinearity (tandem duplications) between genes located on different chromosomes. Chromosome numbers are shown in grey inside the chromosome blocks (numbers 1–17).

**Figure 7 plants-12-02349-f007:**
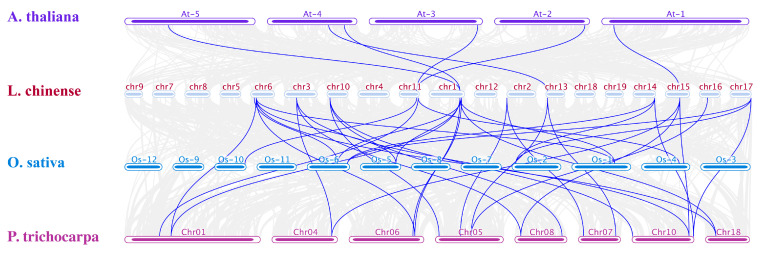
Interspecies GATA gene collinearity. GATA gene collinearity analysis between *L. chinense* (chromosomes are depicted with maroon blocks) and *A. thaliana* (violet blocks), *O. sativa* (blue blocks), and *P. trichocarpa* (purple blocks). Blue curvy lines show collinear GATA genes between the four plant species, each chromosome was named above.

**Figure 8 plants-12-02349-f008:**
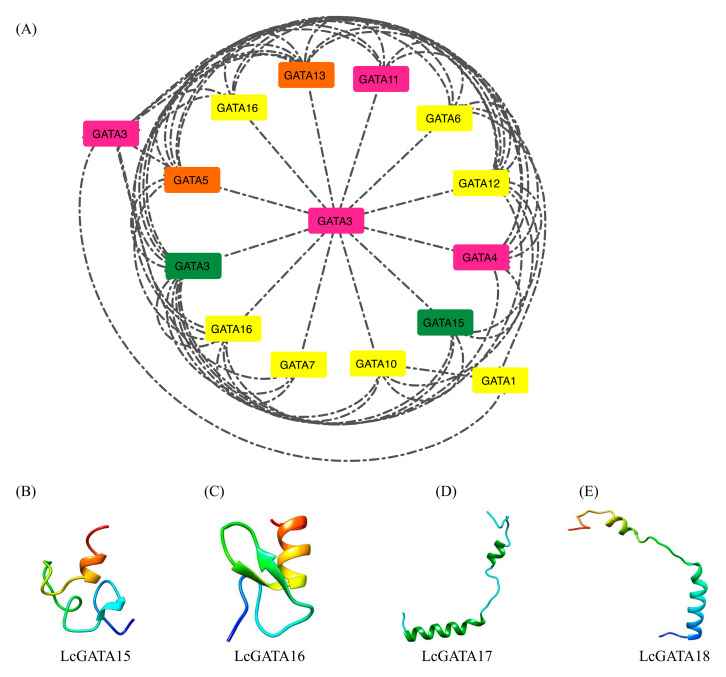
Protein interaction and structure analysis. (**A**) Protein-protein interaction network for LcGATAs was analyzed using the STRING website (http://string-db.org, accessed on 17 June 2022) using the full-length protein sequences of the LcGATA family. *Arabidopsis thaliana* was used as a reference plant species. Different color schemes show different evolutionary LcGATA groups: Group A (green), B (orange), C (yellow), and D (purple). (**B**–**E**) The 3D protein structure prediction of LcGATA15/16/17/18, respectively, showing the potential strands, helices, and coil formation.

**Figure 9 plants-12-02349-f009:**
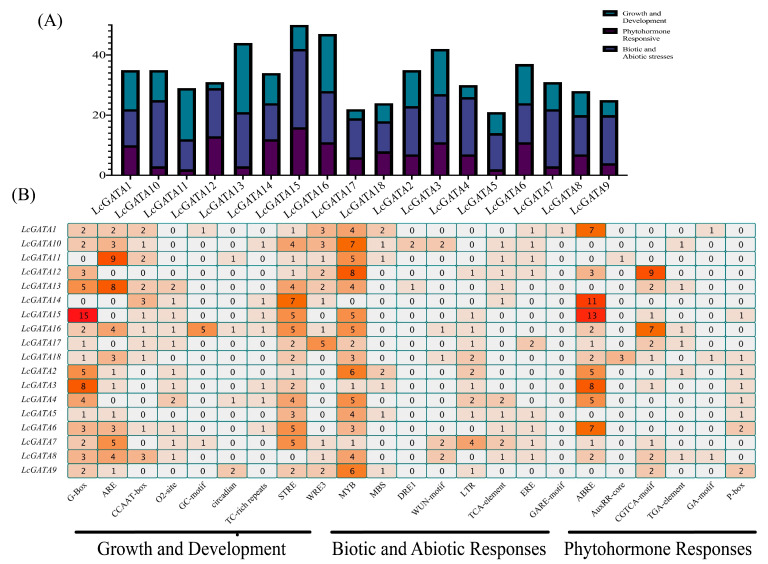
*Cis-regulatory* element analysis. (**A**) The total numbers of the identified putative *cis*-elements in the promoter of *L. chinense* GATA genes, shown as a bar graph. Different color schemes are described in the key (top-right). (**B**) Detailed *cis*-regulatory element representation in various promoter regions of the LcGATA genes, presented using a heatmap. Different color backgrounds show the total ranges of *cis*-elements present.

**Figure 10 plants-12-02349-f010:**
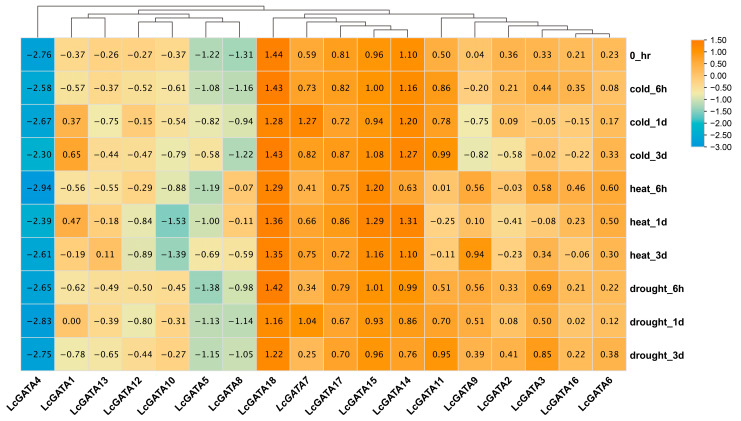
Transcriptome expression analysis. A heatmap showing eighteen LcGATA gene responses to three abiotic stresses (cold, heat, and drought), generated by the Tbtools software (https://bio.tools/tbtools, accessed on 23 February 2023). The expression levels were recorded at different time points, 0 h (control), 6 h, 1 day (1 d), and 3 d, shown in bold after the stress on the right side. The GATA genes investigated are depicted at the bottom. The expression levels of each gene were normalized to the maximum value, and values were expressed in the log and negative form (key shown in the right corner).

**Figure 11 plants-12-02349-f011:**
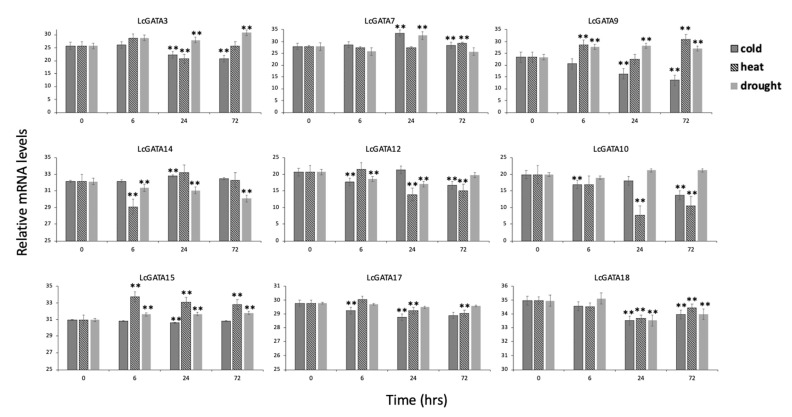
qPCR gene expression analysis of 9 LcGATA genes in different abiotic stresses (cold, heat, and cold) at different time points (6 h, 24 h, and 3 d). The relative mRNA levels are depicted in the *y*-axis over time in the *x*-axis. Significant differences are shown above the bar graphs using double asterisks (**). Error bars were generated using the standard error value calculated by Microsoft Excel.

**Table 1 plants-12-02349-t001:** Physicochemical properties of LcGATA members.

Gene ID	Gene Name	Chromosome	Chromosome Location	aa	Mw(KDa)	pI	Localization
Lchi11034	LcGATA1	Chr 1	76673911; 76676124	327	36.0	5.53	Nucleus
Lchi00930	LcGATA2	Chr 1	97844054; 97846142	321	35.5	6.76	Nucleus
Lchi00761	LcGATA3	Chr 1	103142854; 103144266	297	32.7	9.34	Nucleus
Lchi33577	LcGATA4	Chr 2	27371; 35262	135	15.1	5.25	Nucleus
Lchi01856	LcGATA5	Chr 2	76545610; 76556256	393	43.5	8.4	Nucleus
Lchi24152	LcGATA6	Chr 3	27419892; 27425752	222	25.2	6.15	Nucleus
Lchi05395	LcGATA7	Chr 3	38862093; 38864782	330	35.4	5.67	Nucleus
Lchi11226	LcGATA8	Chr 4	38490148; 38502630	366	41.0	4.63	Cytoplasm/Nucleus
Lchi03424	LcGATA9	Chr 6	11359264; 11359944	199	22.4	9.94	Nucleus
Lchi08934	LcGATA10	Chr 6	17595500; 17597071	343	37.2	5.83	Nucleus
Lchi30686	LcGATA11	Chr 10	9467120; 9467702	120	13.4	9.71	Nucleus
Lchi01461	LcGATA12	Chr 11	55131852; 55133154	268	29.1	6.30	Nucleus
Lchi01268	LcGATA13	Chr 11	59911870; 59920955	583	64.3	6.93	Nucleus
Lchi13044	LcGATA14	Chr 13	5648353; 5661943	624	67.7	6.1	Nucleus
Lchi03334	LcGATA15	Chr 14	68497886; 68527212	504	55.9	9.81	Nucleus
Lchi08164	LcGATA16	Chr 15	45414499; 45425579	289	31.7	10.05	Nucleus
Lchi17957	LcGATA17	Chr 16	25722343; 25736319	339	36.7	5.83	Nucleus
Lchi05152	LcGATA18	Chr 17	66662271; 66670843	299	33.0	8.33	Nucleus

**Table 2 plants-12-02349-t002:** The Ka/Ks values of LcGATA linked genes.

SEQ_1	SEQ_2	KA	KS	KA_KS
LCGATA10	LcGATA2	0.17044698	0.85528245	0.19928735
LCGATA16	LcGATA7	0.25063103	1.27657256	0.19633121
LCGATA18	LcGATA15	0.3122753	1.03194528	0.30260839
LCGATA13	LcGATA5	0.45604965	2.18390847	0.20882269
LCGATA17	LcGATA14	0.35202336	1.50301022	0.23421222

## Data Availability

Genome and gene model annotations files are available on the NCBI website (https://www.ncbi.nlm.nih.gov/assembly/GCA_003013855.2, accessed on 15 June 2022). Transcriptome datasets are also available on the NCBI website; the cold and heat stress accession number is PRJNA679089 (https://www.ncbi.nlm.nih.gov/bioproject/PRJNA679089/, accessed on 15 June 2022), and the drought stress accession number is PRJNA679101 (https://www.ncbi.nlm.nih.gov/bioproject/PRJNA679101/, accessed on 15 June 2022).

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
