# Peer review of "Systematic Characterization of GATA Transcription Factors in *Liriodendron chinense* and Functional Validation in Abiotic Stresses"

_plants, 2023, doi:10.3390/plants12122349_

Round 1

Reviewer 1 Report

I think the manuscript presents interesting and novel data however it needs to be thoroughly revised before it will be acceptable for publication in Plants.

Below I list the detailed points that I consider need attention.

1. The English style must be improved since in some sections the manuscript is difficult to follow and there are also many grammatical and spelling errors throughout.

2.  In Figure 3, the dendrogram in part A needs to be improved, the lettering is small and difficult to read even when increased in size. The authors should comment in the results and in more detail in the discussion section the significance of the different groups, do they have properties, expression patterns etc in common.

3.  Figure 3 part B is really a comparison between families not species and it is not clear what the colors represent in the table in part C.

4. I think there is a mistake in Figure 4, parts B and D seem to be mixed up and I think both sections basically show the same result. I think inclusion of the example in only two colors, yellow and green would be enough.

5. I think from Figure 2 and Figure 5 A there is evidence for some tandem duplication of genes or perhaps I misunderstood the data. This is not mentioned in results or discussed.

I think it is not correct to carry out analysis of synteny between species which are taxonomically so far apart-has this been done before can the authors show some support or cite other publications where this is done routinely?

6. I do not understand the data presented in Figure 6 Why do proteins with similar function have such different structures? Why were they tested for interactions with proteins from Arabidopsis, do they correspond to expected structures from other plant species? These points should all be covered also in the discussion section.

7. There are 2 Figure 8s. In the second (Figure 9?) it would be better to present the data comparing changes within the same stress condition at different time points, not different stresses at the same time point. It is very confusing and hard to determine whether expression changes over time for each stress.

8. The discussion section needs to clearly show the significance of the results in comparison to similar studies in other species and to comment in detail on the similarities or differences observed.

Author Response

Response to Reviewer 1 Comments

Dear Reviewer and Editor, 

Thank you very much for your time and efforts in handling and reviewing our manuscript. Your valuable comments and suggestions helped us significantly improve the quality of the manuscript. We have revised the manuscript according to your insightful comments and suggestions and tried to improve the quality of the manuscript. And point-to-point responses were made and listed below.

Comment 1. The English style must be improved since in some sections the manuscript is difficult to follow and there are also many grammatical and spelling errors throughout.

Response: We apologize for the manuscript state in our initial submission. However, all the grammatical mistakes, spelling errors, and English proficiency have been thoroughly revised with professional assistance. Please see the manuscript. 

Comment 2.  In Figure 3, the dendrogram in part A needs to be improved, the lettering is small and difficult to read even when increased in size. The authors should comment in the results and in more detail in the discussion section the significance of the different groups, do they have properties, expression patterns etc in common.

Response 2: Thank you for this comment. Figure 3 was corrected, and the dendrogram was enlarged. For presentation purposes, Figure 3A was displayed separately and now Figures 3B and C have been shown as Figure 4. We hope this edition will have a better display. In addition, the result comment and discussion of these Figures were enhanced to provide full detail and   the significance of the different plant groups, their properties and expression patterns. Please see Subsection 3.1.

  1. Figure 3 part B is really a comparison between families not species and it is not clear what the colors represent in the table in part C.

Response 3: We apologize for this oversight, and Figure 4 indeed shows a comparison of different plant orders and their LcGATA transcription factor presentation. The various color schemes in Figure 4C show a heatmap depicting ranges of the total GATA transcription factors in each plant. Colors green to red represent the least to the most GATA TFs.

  1. I think there is a mistake in Figure 4, parts B and D seem to be mixed up and I think both sections basically show the same result. I think inclusion of the example in only two colors, yellow and green would be enough.

Response 4: We greatly value this observation, and please accept our apologies for the caption mistake. Figure 5B shows the protein motif arrangement in each identified LcGATA protein. Therefore, different color schemes depict identified motifs (1-14) denoted by numbers. On the other hand, Figure 5C shows the exon-intron arrangements (LcGATA gene structures), represented by two colors, green-yellow, respectively. Nonetheless, we have corrected our Figure caption and included all this information. Please see Line 811-818.  

  1. I think from Figure 2 and Figure 5 A there is evidence for some tandem duplication of genes, or perhaps I misunderstood the data. This is not mentioned in results or discussed.

I think it is not correct to carry out analysis of synteny between species which are taxonomically so far apart-has this been done before can the authors show some support or cite other publications where this is done routinely?

Response 5: Thank you for this analysis. However, according to our results, the obtained duplicated genes were not inserted at the same loci or original DNA segment (tandem duplication). But some duplicates were on different loci (segmental duplication) although on the same chromosome, providing evidence that these were segmental duplication events. We hope this short explanation will befit this question.

Regarding the synteny analysis, we performed a comparison with O. sativa and A. thaliana since in these plants much research has been done as compared to plant species related to the L. chinense; which might give a hint of possible functions in the L. chinense GATA genes. Similar work has been done in the following:

  1. Li, M.; Hwarari, D.; Li, Y.; Ahmad, B.; Min, T.; Zhang, W. The bZIP transcription factors in Liriodendron chinense: Genome-wide recognition, characteristics, and cold stress response. Frontiers in Plant Sci. 2022, article 13.
  2. Guan, Y.; Liu, S.; Wu, W.; Hong, K.; Li, R.; Zhu, L. Genome-wide identification and cold stress-induced expression analysis of the CBF gene family in Liriodendron chinense. Journal of Fores Res. 2021, 32(6), 2531-2543.
  3. Hwarari, D.; Guan, Y.; Li, R.; Movahedi, A.; Chen, J.; Yang, L. Comprehensive Bioinformatics and Expression Analysis of TCP Transcription Factors in Liriodendron chinense Reveals Putative Abiotic Stress Regulatory Roles. Forests. 2022, 13(9), article 1401.
  4. Ke, Y.; Xu, M.; Hwarari, D.; Ahmad, B.; Li, R.; Guan, Y. OSCA Genes in Liriodendron chinense: Characterization, Evolution, and Response to Abiotic Stress. Forests. 2022, 13(11), articled 1835.
  5. Li, R.; Radani, Y.; Ahmad, B,.; Movahedi, A.; Yang, L. Identification and characteristics of SnRK genes and cold stress-induced expression profiles in Liriodendron chinense. BMC genomics. 2022, 23(1), 1-18.

  1. I do not understand the data presented in Figure 6 Why do proteins with similar function have such different structures? Why were they tested for interactions with proteins from Arabidopsis, do they correspond to expected structures from other plant species? These points should all be covered also in the discussion section.

Response 6: We apologize for an inadequate explanation of these Figures, and value this observation. The LcGATA proteins were investigated for 3D protein structures. Our findings showed varied structures, confirming the range of their functions. Additionally, these proteins were examined for protein network using the A. thaliana as a reference plant since A. thaliana is a model plant and the L. chinense data is not yet available on the STRING database. We have provided a complete description of Figure 7 in the manuscript and Discussion section. Please see Line 173-191 and the Discussion section.

  1. There are 2 Figure 8s. In the second (Figure 9?), it would be better to present the data comparing changes within the same stress condition at different time points, not different stresses at the same time point. It is very confusing and hard to determine whether expression changes over time for each stress.

Response 7: Thank you for this comment. Figure numbering was corrected as advised. In Figure 10, the heat map shows the LcGATA gene response at different time points (after 6hrs, 24hrs, and 3 days), compared in three various stresses (cold, heat, and drought). The bottom panel depicts all the time points and the stress (for example, heat_1d represents heat stress assessed after 1 day.).

  1. The discussion section needs to clearly show the significance of the results compared to similar studies in other species and comment in detail on the similarities or differences observed.

Response 8: Thank you for this comment. The discussion section was further enlarged, and comments regarding similar studies in other species were thoroughly deliberated, pointing out their similarities and differences. Please see the Discussion section.

Reviewer 2 Report

 GATA transcrip- 11 tion factors (TFs) were reported to play a vital role in response to abiotic stress in plants. In this 12 study, 18 GATA genes were identified in the genome of L. chinense, the identified genes clustered 13 together in 4 separate groups according to phylogenetic relationship, gene structure and arrange- 14 ment, and motif conservation. they the identified LcGATA genes 21 demonstrated an abundance of cis-regulatory elements, especially those of the abiotic stress re- 22 sponses. Both the transcriptomic and qPCR gene expression analyses confirmed the involvement of 23 LcGATAs in various stresses including heat, cold and drought stress. Particularly, LcGATA15, 24 LcGATA17, and LcGATA18, were highly upregulated in all abiotic stresses.

the novelty of this work is not high and can be improved

i still have have some comments

first how you did the stress conditions

in the introduction part i need to see the relationship between you work and different stress conditions

your figs is not clear

make deep disc

Author Response

Response to Reviewer 2 Comments

Dear Reviewer and Editor, 

Thank you very much for your time and efforts in handling and reviewing our manuscript. Your valuable comments and suggestions helped us significantly improve the quality of the manuscript. We have revised the manuscript according to your insightful comments and suggestions and tried to improve the quality of the manuscript. And point-to-point responses were made and listed below.

Comment 1. the novelty of this work is not high and can be improved

Response 1: We value this comment and have taken the time to enhance the originality of the research. Similar studies were also deliberated on, and additional analysis was done pointing out the relevance of this research in plant genomics and evolution.

Comment 2. i still have have some comments, first how you did the stress conditions

Response 2: Thank you for this question. The procedure for abiotic stressing was done as follows;

The L. chinense somatic embryo-fetal regenerated seedlings from a single batch of constant growth were separated into three different incubators with different treatments; 4 OC for the cold stress treatment, 35-40 OC for the heat treatment, and 40% polyethylene glycol/PEG6000 for the drought stress. Mature leaves were collected at 0h (control), 6h, 1d, and 3d post-treatment and then stored at −80 ◦C for further use [22].

Please see the manuscript, Line 290-293.

Comment 3. in the introduction part i need to see the relationship between you work and different stress conditions.

Response 3: We value this suggestion and comment. Nonetheless, the introduction section was enhanced, and parallels were drawn to show the relationship between this research and different stress conditions. Please see Lines 109-139.

Comment 4. your figs is not clear

Response 4: We apologize for this oversight. All the figures were enhanced, and some were redrawn. Please see the manuscript, we hope the corrected figures will satisfy you.

Comment 5. make deep disc

Response 5: Thank you for this comment. The discussion section was thoroughly revised, and comments regarding similar studies in other species were thoroughly deliberated, pointing out their similarities and differences to this study. We hope this revised discussion section will be to your satisfaction. Please see the Discussion section.

Reviewer 3 Report

In this manuscript, the authors investigated the GATA transcription factor family in the genome of Liriodendron Chinese. The authors studied this gene family in the ways of phylogenetic analysis, gene structure and motif analyses. Further transcriptional analysis were carried out to examine the expression pattern of these genes in response to abiotic stress via transcriptome and qPCR. The authors observed three genes are unregulated in response to abiotic stresses, concluding pivotal roles of these GATA genes in regulating abiotic stress. However, more experimental evidence should be provided to support the potential roles of GATAs as the authors proposed. My comments are listed in detail as follows.

1. In introduction, more detailed information of the mechanisms of GATA involved in abiotic stress response should be provided.

2. In the Materials and Methods, more method information about the RNAseq  should be provided in the manuscript.

3. What is the possible reasons of fewer GATA genes in L. Chinese than those in Arabidopsis and rice.

4. Figure 1, are these protein sequences zinc-finger domains? If so, please describe it in the figure legend

5. Transcriptional correlation with CBF is not strong enough to support the conclusion that GATA are involved in stress  response

Author Response

Response to Reviewer 3 Comments

Dear Reviewer and Editor,

Thank you very much for your time and efforts in handling and reviewing our manuscript. Your valuable comments and suggestions helped us greatly improve the manuscript's quality. We revised the manuscript according to your factful and valuable comments and suggestions, and tried to improve the quality of the manuscript, making point-to-point response as follows;

Point 1. In the introduction, more detailed information of the mechanisms of GATA involved in abiotic stress response should be provided.

Response 1

Thank you for this suggestion and comment. Supporting references of the involvement of GATA genes in abiotic stress responses have been added to the manuscript. Please see the Introduction section.

Point 2. In the Materials and Methods, more method information about the RNAseq should be provided in the manuscript.

Response 2

Thank you very much for this suggestion and comment. Additional regarding the RNA seq has been added in the Materials and Methods section. Please see the manuscript.

Point 3. What is the possible reasons of fewer GATA genes in L. Chinese than those in Arabidopsis and rice.

Response 3

Thank you for this question, we value it very much. We speculate that the possible reason for fewer GATA genes in L. Chinese than those in Arabidopsis and rice may be due to:

  1. Gene family size largely depends on the gene duplication events such as tandem duplication and segmental duplication. Thus, possibly the degree and extent of duplication in the LcGATA gene family were lesser than those in the Arabidopsis and rice.
  2. Plants often evolve specialized adaptations to their local environments, such as drought tolerance or disease resistance. Adaptation strategies in the Arabidopsis and rice possibly led to the increased number of GATA gene families.

In addition, we apologize for not including adequate information regarding this question. However, this information was added in the discussion section.

Point 4. Figure 1, are these protein sequences zinc-finger domains? If so, please describe it in the figure legend

Response 4

Thank you very much for your question and suggestion. The logos show protein zinc-finger domains, and this information was added in the Figure legend as advised.

Point 5. Transcriptional correlation with CBF is not strong enough to support the conclusion that GATAs are involved in stress responses.

Response 5

We value this comment and thank you very much. The correlation analysis showed similar expression trends between the LcCBFs and the LcGATAs in different abiotic stress, suggesting a possible interconnection reaction. However, this result does not provide substantial evidence and cannot be the only analysis to prove the involvement of these LcGATAs in stress responses. This information was also added to the manuscript as advised. 

If this explanation is insufficient, the content of this section may be deleted from the manuscript.

Reviewer 4 Report

In this manuscript, the authors explored the GATA transcription factors in Liriodendron chinense genome studying their functionality under abiotic stresses. Some points need to be corrected and/or clarified.

Revise along all manuscript the organism scientific name “Liriodendron chinense”. Some times the authors write the species name as chinese instead chinense. Also, avoids to use the species name in capital “Chinense”.

The first 4 lines of abstract need to be rewritten.

Line 29: Use point instead comma, and continue with capital letter the conclusion.

Line 38 After … ”perennial habit” it is necessary a reference.

Lines  53 and 54. The authors open parentheses but not close them…

Line 84. Remove “Although,”

Lines 86 to 89 need to be revised.

Line 102 and 193 correct physiochemical to physicochemical

Line 104. I do not understand how a genomic sequence can be obtained from a protein database. Also, it is necessary to include the link of the database site.

Line 123 and 124. Includes a reference for MEGA 11 software.

Line 129. I would like to see the protein subcellular localization using the DeepLoc-2.0 tool https://services.healthtech.dtu.dk/services/DeepLoc-2.0/. It uses machine learn and could give a more precise localization.

For qRT-PCR analysis:

- Have the authors tested the genomic DNA contamination in RNA samples? 

- About the primers. Have the authors obtained the annealing temperatures experimentally? Primer pair efficiencies were performed?

- Also, It is also very important to verify the primer specificity experimentally. On this, were melting curve performed? Or were PCR products visualized in a agarose gel?

Line 185. “LcGATA proteins” instead “LcGATA genes”

Lines 209, 266 and 299. Use comma instead point, and continue with lower case

Includes reference to Xshell software and the online tool (iTOL)

Figure 7A is very difficult to understand. I think it is necessary to explain the colors and / or to include letters representing species as At for Arabidopsis or Lc to Liriodendron chinense. 

Line 380. “This result affirms that the GATA genes in L. chinense are engaged in a wide range of biological functions, including abiotic stress response.” I think it is better to use “indicates” instead “affirms” since this result was derived from in silico analysis.

Lines 406 and 538. “stresses” instead “stress”

Figure 9. I don’t understand the color scale varying from 1.5 to -3.0 when there are numbers for gene expression varying from 0 to more than 11. Other question, have the authors applied statistical analysis on these data to be sure about the up and down regulations described in the results? Lines 396 and 397 authors state that “there was an insignificant downregulation expression pattern in LcGATA11 and LcGATA14”, but no statistical analyses was informed. In addition, I think it is important to inform in the legend if these data were obtained in relation to control conditions. What represent these numbers in relation to control values (each time have a control?)? These informations need to be included in the material and methods. It is very confuse in the present form. Have the transcriptome data published in a public database? In case positive, please inform in material the accession number.

Lines 392 to 393. Change the words “two genes do not respond” to “two genes are repressed”

Figure 10. Why the authors not included the time 0h? I don’t understand these results. How the authors have normalized these data?

I think that the expression data in Figures 9 and 10 need to be revised carefully.

Line 439, remove “and” before the name Arabidopsis.

Line 556. “Data Availability Statement: Not Applicable” What about the transcriptomic data?

Author Response

Response to Reviewer 4 Comments

Dear Reviewer and Editor,

Thank you very much for your time and efforts in handling and reviewing our manuscript. Your valuable comments and suggestions helped us greatly to improve the quality of the manuscript. We revised the manuscript according to your factful and valuable comments and suggestions, and tried to improve the quality of the manuscript, making point-to-point response as follows;

Point 1. Revise along all manuscript the organism scientific name “Liriodendron chinense”. Some times the authors write the species name as chinese instead chinense. Also, avoids to use the species name in capital “Chinense”.

Response 1: Thank you for this comment and observation. The scientific name “Liriodendron chinense” was corrected throughout the manuscript, and the capital “C” in chinense was changed to a small letter.

Point 2. The first 4 lines of abstract need to be rewritten.

Response 2: We value this comment and contribution, the first four lines and the whole Abstract were rewritten. Please see the Abstract section.

Point 3. Line 29: Use point instead comma, and continue with capital letter the conclusion.

Response 3: We are grateful for this comment, the comma in line 29 was replaced with a point as advised. Please see now Line 27.

Point 4. Line 38 After … ”perennial habit” it is necessary a reference.

Response 4: Thank you very much for this observation and suggestion. Two references were added after “perennial habit [1,2]” which are:

  1. Cao, Y.; Feng, J.; Hwarari, D.; Ahmad, B.; Wu, H.; Chen. Alterations in Population Distribution of Liriodendron chinense (Hemsl.) Sarg. and Liriodendron tulipifera Caused by Climate Change. Forests. 2022, 13(3), article 488.
  2. Parks, C.R.; Wendel, J.F. Molecular divergence between Asian and North American species of Liriodendron (Magnoliaceae) with implications for interpretation of fossil floras. Amer Jour Bot. 1990, 77(10), 1243-1256.

Point 5. Lines  53 and 54. The authors open parentheses but not close them…

Response 5: We apologize for this error, and appreciate this observation. Line 53 and 54 parentheses were closed. Please see Lines 53 and 54.

Point 6. Line 84. Remove “Although,”

Response 6: Thank you for this suggestion. The word “Although” was removed as advised, please see Line.

Point 7. Lines 86 to 89 need to be revised.

Response 7: Thank you very much for this comment. Lines 86 to 89 were revised and rewritten. Please see the manuscript.

Point 8. Line 102 and 193 correct physiochemical to physicochemical

Response 8: We apologize for this error. The word “physicochemical” was corrected as advised. Please see the manuscript.

Point 9. Line 104. I do not understand how a genomic sequence can be obtained from a protein database. Also, it is necessary to include the link of the database site.

Response 9: We regret this oversight; the genomic sequences were obtained from the Hardwood genomics database. This information was added to the manuscript.

The genomic and protein sequences of Liriodendron chinense were collected from the hardwood genomics database (now, TreeGenes database: https://treegenesdb.org/) [22]; following previous publications [23, 24] (accessed on the 12th of June 2022),

Point 10. Line 123 and 124. Includes a reference for MEGA 11 software.

Response 10: We value this suggestion, reference for MEGA X software was added as follows;

  1. Kumar, S.; Stecher, G.; Li, M.; Knyaz, C.; Tamura, K. MEGA X: molecular evolutionary genetics analysis across computing platforms. Mol Biol Evol. 2018, 35(6), article.1547.

Point 11. Line 129. I would like to see the protein subcellular localization using the DeepLoc-2.0 tool https://services.healthtech.dtu.dk/services/DeepLoc-2.0/. It uses machine learn and could give a more precise localization.

Response 11: Thank you for this suggestion. Below attached are results based on the DeepLoc-2.0. In addition, these results were also used in the manuscript.

Protein_ID

Localizations

Signals

Cytoplasm

Nucleus

Extracellular

Cell membrane

Mitochondrion

Plastid

Endoplasmic reticulum

Lysosome/Vacuole

Golgi apparatus

Peroxisome

LcGATA7

Nucleus

Nuclear localization signal

0.1991

0.9619

0.0052

0.0284

0.0576

0.0486

0.0308

0.0351

0.0091

0.0087

LcGATA1

Nucleus

Nuclear localization signal

0.1758

0.94840002

0.0048

0.0342

0.0515

0.0545

0.0319

0.0373

0.0096

0.0096

LcGATA9

Nucleus

Nuclear localization signal

0.2262

0.95740002

0.0106

0.0573

0.1149

0.0767

0.0404

0.009

0.0107

0.0228

LcGATA4

Nucleus

0.2965

0.81269997

0.0375

0.0269

0.15719999

0.12720001

0.1392

0.0736

0.0694

0.0132

LcGATA13

Nucleus

Nuclear localization signal

0.1895

0.9465

0.0123

0.0537

0.0331

0.0546

0.0459

0.0259

0.0156

0.0237

LcGATA12

Nucleus

Nuclear localization signal

0.1645

0.95459998

0.0093

0.0496

0.0523

0.0513

0.0302

0.0497

0.0104

0.0106

LcGATA8

Cytoplasm|Nucleus

Nuclear localization signal

0.53219998

0.64490002

0.0922

0.13699999

0.0471

0.0367

0.156

0.1698

0.0871

0.0226

LcGATA16

Nucleus

Nuclear localization signal

0.1965

0.96270001

0.0083

0.0256

0.0536

0.0599

0.0299

0.0275

0.0094

0.0156

LcGATA3

Nucleus

Nuclear localization signal|Nuclear export signal

0.23909999

0.93180001

0.009

0.0672

0.1036

0.0708

0.0429

0.0167

0.0142

0.0328

LcGATA2

Nucleus

Nuclear localization signal

0.16859999

0.94019997

0.0129

0.0462

0.0647

0.0563

0.0238

0.0217

0.0077

0.0154

LcGATA15

Nucleus

Nuclear localization signal

0.2361

0.85140002

0.057

0.1111

0.0583

0.0587

0.0923

0.0504

0.0379

0.0012

LcGATA6

Nucleus

Nuclear localization signal

0.1684

0.97390002

0.0047

0.0361

0.0672

0.032

0.0374

0.0265

0.0053

0.0088

LcGATA14

Nucleus

Nuclear localization signal

0.1804

0.95740002

0.0134

0.0553

0.0764

0.0498

0.0481

0.0666

0.0208

0.0054

LcGATA5

Nucleus

Nuclear localization signal

0.2005

0.95999998

0.0094

0.0377

0.0483

0.132

0.0334

0.0231

0.022

0.0518

LcGATA17

Nucleus

Nuclear localization signal

0.2156

0.93379998

0.0226

0.0827

0.09

0.04

0.0467

0.0822

0.0469

0.0021

LcGATA11

Nucleus

Nuclear localization signal|Nuclear export signal

0.28830001

0.92619997

0.0081

0.07

0.0981

0.081

0.0756

0.0069

0.0248

0.0327

LcGATA10

Nucleus

Nuclear localization signal

0.1486

0.94779998

0.0076

0.0485

0.0491

0.0501

0.0278

0.0365

0.0114

0.0207

LcGATA18

Nucleus

Nuclear localization signal

0.2392

0.91540003

0.0189

0.0799

0.0975

0.0419

0.0488

0.0691

0.0397

0.0014

Point 12. For qRT-PCR analysis:

- Have the authors tested the genomic DNA contamination in RNA samples?

Response 12: We value this question so much, The KK-rapid plant total RNA extraction kit used in qPCR also contains a radical group removal module, DNAse (+gDNA wiper) as one of its advantages; that actively removes upto 500ng of radical DNA effectively. So yes in this experiment a DNA contamination removal in RNA samples was performed.

Point 13. - About the primers. Have the authors obtained the annealing temperatures experimentally? Primer pair efficiencies were performed?

Response 13: Thank you very much for this question, in this experiment we verified our primer's annealing temperatures using the Gradient PCR method. The range of LcGATA primer annealing temperatures was broadened and it allowed the determination of the temperature at which the primers perform best in terms of specificity and efficiency.

Point 14. - Also, It is also very important to verify the primer specificity experimentally. On this, were melting curve performed? Or were PCR products visualized in an agarose gel?

Response 14: Thank you very much for this question, and yes the primers were verified for specificity through visualization of the PCR products in an agarose gel. Below attached is the result from the Gel electrophoresis.

Point 15. Line 185. “LcGATA proteins” instead “LcGATA genes”

Response 15: We apologize for this error; the word "genes" was replaced with protein as advised. Please see the manuscript.

Point 14. Lines 209, 266 and 299. Use comma instead point, and continue with lowercase

Response 14: We are really thankful for this insight. Points were replaced with a comma and lower case format was maintained. Please see the manuscript.

Point 15. Includes reference to Xshell software and the online tool (iTOL)

Response 16: Thank you very much for this advice. The corresponding references were added to the manuscript as:

  1. Emms, D.M.; Kelly, S. "OrthoFinder: phylogenetic orthology inference for comparative genomics." Genome biology20. 2019, 1-14.
  2. Letunic, I.; Bork, P., 2019. Interactive Tree Of Life (iTOL) v4: recent updates and new developments. Nucleic acids research, 47(W1), pp.W256-W259.

Point 17. Figure 7A is very difficult to understand. I think it is necessary to explain the colors and / or to include letters representing species as At for Arabidopsis or Lc to Liriodendron chinense. 

Response 17: We apologize for the state of Figure 7A in our initial submission. Nonetheless, we have redrawn the figure and included additional GATA proteins that exist at lower threshold values. In detail, the now Figure 8A shows a protein-to-protein interaction of GATA proteins using an orthologous protein of Arabidopsis to L. chinense. In addition, these proteins were renamed according to their respective protein in L. chinense. Different color schemes and shapes represent different LcGATA groups. All this information has been added to both the Figure legend and the manuscript content. 

Point 18. Line 380. “This result affirms that the GATA genes in L. chinense are engaged in a wide range of biological functions, including abiotic stress response.” I think it is better to use “indicates” instead “affirms” since this result was derived from in silico analysis.

Response 18: Thank you very much for this suggestion, the word affirms was changed to indicates. Please see Line 380.

Point 19. Lines 406 and 538. “stresses” instead “stress”

Response 19: We value this suggestion and comment. The word stresses was added instead of stress. Please see Lines

Point 20. Figure 9. I don’t understand the color scale varying from 1.5 to -3.0 when there are numbers for gene expression varying from 0 to more than 11. Other question, have the authors applied statistical analysis on these data to be sure about the up and down regulations described in the results? Lines 396 and 397 authors state that “there was an insignificant downregulation expression pattern in LcGATA11 and LcGATA14”, but no statistical analyses was informed. In addition, I think it is important to inform in the legend if these data were obtained in relation to control conditions. What represent these numbers in relation to control values (each time have a control?)? These informations need to be included in the material and methods. It is very confuse in the present form. Have the transcriptome data published in a public database? In case positive, please inform in material the accession number.

Response 20: We value this point and suggestions, and our deepest regret on the state of Figure 9 and its inadequacies. However, we have redrawn Figure 10 and changed the Figure legend according to the gene expression values. All the gene expression values transformed the log base 2, and later normalized to the maximum value (1.5). Further Tbtools software was used to generate a heatmap for presentation purposes. Regarding the statistical analysis, no statistical package or parameters were used in the analysis of this data, hence its presentation was totally based on the data depictions. In addition, all this information was added to the Figure legend and included in the Materials and Methods section. The transcriptome data was published and readily available. Please see manuscript

Point 21. Lines 392 to 393. Change the words “two genes do not respond” to “two genes are repressed”

Response 21: Thank you for this suggestion, the word phrase was edited as advised. Please see Lines 392 to 393. 

Point 22. Figure 10. Why the authors not included the time 0h? I don’t understand these results. How the authors have normalized these data?

Response 22: We apologize for the oversight in Figure 10 of our initial submission. We have redrawn figure 10 and included mRNA levels at 0hr. In addition, we have expanded the result analysis and explanation, we hope this will be to your satisfaction. 

Point 23. I think that the expression data in Figures 9 and 10 need to be revised carefully.

Response 23: Thank you for this suggestion, the expression data in Figures 9 and 10 were revised according and the Figures were redrawn to depict the intended information. Please see Figures 9 and 10 in the manuscript.

Point 24. Line 439, remove “and” before the name Arabidopsis.

Response 24: We apologize for this oversight, the word “and” was removed as advised. Please see Line

Point 25. Line 556. “Data Availability Statement: Not Applicable” What about the transcriptomic data?

Response 25: We apologize for this omission. The “Data Availability Statement” was added to the manuscript. Please see the manuscript.

Round 2

Reviewer 2 Report

the authors is working about The GATA transcription factors 11 (TFs) were reported to play a vital role in response to abiotic stress in plants. In this study, 18 GATA 12 genes were identified in the genome of L. chinense, the identified genes clustered together in 4 13 separate groups according to phylogenetic relationship, gene structure and arrangement, and motif 14 conservation. the idea is ok

but i still have comments 

English writing need strong improvement

plz rewrite your Abstract

the introduction must be clear in particular the aim of your work

improve your figs quality

deep discussion

Author Response

Response to Reviewer 2 Comments (Round 2)

Dear Reviewer and Editor, 

Thank you very much for your time and efforts in handling and reviewing our manuscript. Your valuable comments and suggestions helped us significantly improve the quality of the manuscript. We have revised the manuscript according to your insightful comments and suggestions and tried to improve the quality of the manuscript. And point-to-point responses were made and listed below.

Comment 1. English writing need strong improvement

Response 1. We apologize for the poor English presentation of our MS. However, we have tried to make strong English improvements, correcting grammar, spelling errors, and content delivery. Thank you for this comment. Please see the manuscript.

Comment 2. plz rewrite your Abstract

Response 2. We value this comment and suggestion. The abstract was re-written and English expression was improved, intended content delivery was enhanced. Please see the abstract section.

Comment 3. the introduction must be clear in particular the aim of your work

Response 3. We deeply regret the state of our introduction. However, the whole section was rewritten to highlight the present research giving a parallel to other previous research. We hope this revised version will be satisfactory.    

Comment 4. improve your figs quality

Response 4. Thank you for this valuable suggestion and comment, all the Figures in the manuscript were enhanced using Adobe Illustrator and saved in jpg. Format on high resolution. We hope the figures are clear now.

Comment 5. deep discussion

Response 5. We value this comment, it improves the quality of our manuscript. The Discussion section was deeply enhanced as advised, additional conclusions were made based on the study findings and parallels were also drawn from other publications. Please see the Discussion section. 

Reviewer 3 Report

no further comments

Author Response

Response to the Editor

Dear Reviewer,

Point 1

no further comments

Response 1

Thank you very much for your time and effort in handling and reviewing our manuscript. Your valuable comments and suggestions helped us greatly improve the manuscript's quality. We revised the manuscript according to your factful and valuable comments and suggestions and tried to improve the quality of the manuscript.

Round 3

Reviewer 2 Report

the authors did all suggestion that i asked

Author Response

Response to Reviewer 2 Comments

Dear Reviewer and Editor,

Thank you very much for your time and efforts in handling and reviewing our manuscript. Your valuable comments and suggestions helped us greatly to improve the quality of the manuscript. We revised the manuscript according to your factful and valuable comments and suggestions, and tried to improve the quality of manuscript, making point-to-point response as follows;

Point 1

the authors did all suggestion that i asked

Response 1

We thank the reviewer for the time and effort invested in our manuscript. All suggestions and comments were incorporated into our revised manuscript.
